# A Wooden Carbon-Based Photocatalyst for Water Treatment

**DOI:** 10.3390/ijms25094743

**Published:** 2024-04-26

**Authors:** Chang Zhang, Shangjie Ge-Zhang, Yudong Wang, Hongbo Mu

**Affiliations:** College of Science, Northeast Forestry University, Harbin 150040, China; zc2021@nefu.edu.cn (C.Z.);

**Keywords:** photocatalyst, biochar, sewage treatment, wood sponge, wood

## Abstract

Due to a large number of harmful chemicals flowing into the water source in production and life, the water quality deteriorates, and the use value of water is reduced or lost. Biochar has a strong physical adsorption effect, but it can only separate pollutants from water and cannot eliminate pollutants fundamentally. Photocatalytic degradation technology using photocatalysts uses chemical methods to degrade or mineralize organic pollutants, but it is difficult to recover and reuse. Woody biomass has the advantages of huge reserves, convenient access and a low price. Processing woody biomass into biochar and then combining it with photocatalysts has played a complementary role. In this paper, the shortcomings of a photocatalyst and biochar in water treatment are introduced, respectively, and the advantages of a woody biochar-based photocatalyst made by combining them are summarized. The preparation and assembly methods of the woody biochar-based photocatalyst starting from the preparation of biochar are listed, and the water treatment efficiency of the woody biochar-based photocatalyst using different photocatalysts is listed. Finally, the future development of the woody biochar-based photocatalyst is summarized and prospected.

## 1. Introduction

With the development of world industrialization and population expansion, a large number of organic and inorganic pollutants are released into nature [1,2,3]. Water is a common and extensive pollution subject, and many complex compounds have been detected in water resources [4]. Persistent organic pollutants (POPs) have attracted the attention of researchers because of their high stability, difficult biodegradation, long residual time in the environment, bioaccumulation, persistence and long-range portability [5].

The research work on photocatalytic degradation of organic pollutants has increased significantly in recent years. Photocatalytic substances are a kind of chemical substance that can play a catalytic role under the excitation of photons, and the common light source is sunlight. As a clean and sustainable energy source, sunlight can induce photocatalysis of photocatalytic substances, eliminate persistent organic pollutants and effectively degrade low concentrations of organic pollutants [6]. Photocatalysts include oxides (titanium dioxide, zinc oxide, tin oxide, zirconium dioxide, etc.), sulfides (cadmium sulfide), semiconductors and some silver salts [7]. Most of these catalysts are used in powder form because the powder photocatalyst has a larger specific surface area and can better react with pollutants, but the disadvantage is that the degradation efficiency is low and there is loss [8]. In fact, due to the low coincidence between the wavelength distribution of sunlight radiation and the photon wavelength corresponding to the energy gap of photocatalysts, most of the sunlight is concentrated in the visible range, and some photocatalysts have very limited absorption of sunlight. The TiO_2_ band gap is 3.2 eV, and only a wide band gap reacts to ultraviolet rays, which account for no more than 4% of the solar energy spectrum [9,10,11]. The band gaps of other common photocatalysts, such as ZnO, TaON, G-C_3_N_4_ and SmVO_4_, are only in the range of about 2.3–2.4 eV [12,13,14,15]. In addition, it is worth noting that some photocatalysts are toxic to the human body, so the recovery of the photocatalyst itself is also an urgent task.

Biomass is widely distributed, renewable, low-pollution, carbon-neutral and widely used in various fields [16,17,18,19,20]. It is usually composed of C, O, H, N and other small elements. In an environment with limited oxygen and at a certain temperature (<950 °C), biomass pyrolyzes to form biochar, the most widely known of which is charcoal [21]. In water treatment, biochar has excellent properties such as a large specific surface area, good ion exchange ability, high porosity and rich oxygen functional groups. However, biochar can only separate pollutants from water by adsorption and cannot eliminate pollutants fundamentally. Moreover, it is difficult for biochar to effectively remove pollutants in high-concentration wastewater with complex components, and it cannot meet the needs of selective removal of pollutants.

This review points out the shortcomings of photocatalysts and biochar in treating pollutants in water (Part I) and briefly introduces the characteristics of biochar (Part II). The development of a supported photocatalyst is reviewed (Part III), the advantages of a multi-carrier wood biochar photocatalyst are emphasized and different production methods are compared (Part IV). After that, the limitations of a wood biochar-based photocatalyst are briefly introduced (Part V), and the future development of wood biochar-based photocatalysts is prospected (Part VI).

## 2. Basic Characteristics of Biochar

### 2.1. Definition of Biochar

Biochar is a kind of carbon-rich material that is produced by pyrolysis (anaerobic heating) of biomass [22], such as wood [23], agricultural residues [24] and other organic materials [25]. In recent years, it has attracted great attention because of its potential applications in various fields, including environmental restoration, water treatment, agriculture and energy production [26,27]. When biochar is used for photocatalytic and adsorption applications, the selection of raw materials plays an important role in determining its properties.

### 2.2. Physical and Chemical Properties of Biochar

The physical and chemical properties of biochar play a vital role in its effectiveness in water treatment applications. These characteristics affect the ability of biochar to remove pollutants, improve water quality and enhance the treatment process.

The high surface area and porosity of biochar provide sufficient binding sites for pollutants through adsorption and ion exchange. This feature enables biochar to effectively remove pollutants such as heavy metals, organic compounds and nutrients from water [28,29]. The chemical composition of biochar, including hydroxyl, carboxyl and phenolic groups, will affect its interaction with pollutants in water. These functional groups can form complex bonds with pollutants, thus improving the removal efficiency of pollutants [30]. The pH value and conductivity of biochar affect its adsorption capacity and its interaction with pollutants in water. By adjusting these characteristics, biochar can be customized for specific pollutants and optimized for water treatment performance [21,31]. The particle size and density of biochar will affect its flow dynamics, sedimentation rate and contact time with water during treatment. These physical characteristics affect the efficiency and scalability of the water treatment system based on biochar [32,33]. The stability of biochar in water treatment systems depends on its structural integrity and degradation resistance, which are very important for its long-term performance and reusability. The regeneration method is also important for maintaining the efficacy of biochar in multiple treatment cycles [34].

The optical characteristics of biochar make it a suitable substrate for photocatalytic applications. Biochar has high absorption characteristics in the visible and near-infrared regions of the electromagnetic spectrum [35]. This means that biochar can capture light energy effectively, which is very important for photocatalytic reactions. The low reflectivity of biochar (especially in the visible spectrum) ensures that more light is absorbed rather than reflected back [36]. This characteristic improves the efficiency of the photocatalytic process which depends on light absorption; The band gap energy of biochar determines the minimum energy required to promote electrons from the valence band to the conduction band, and electrons can participate in photocatalytic reaction in the conduction band [37]. A smaller band gap usually corresponds to better photocatalytic activity.

### 2.3. Source of Biochar

Wood is one of the most common sources of biochar [21], especially wood such as balsa wood [38], white oak [39], poplar wood [40] and walnut shell [41]. Wood biochar usually has the characteristics of high carbon content and rich pore structure, which make it perform well in adsorption and catalytic applications [42]. In addition to wood, other biomass materials can also be used to prepare biochar, such as straw [43], rice hull [44], waste crops [45], sugarcane [46], lotus leaf [47] and cellulose matrix [48]. Straw biochar may have a high ash content [49], while shell biochar may contain rich trace elements [50], which can affect its performance and application in photocatalysis and adsorption.

## 3. Woody Biochar-Based Photocatalyst

Coupling the matrix with a photocatalyst is a simple way [51,52] to support the photocatalytic system (Figure 1). Compared with a single nanoparticle, the matrix of the supported photocatalytic system makes the photocatalyst have a higher photodegradation efficiency and recovery rate of organic pollutants, and the types of supported matrices are silica [8,53,54], metal-type [55,56], biochar-type, etc. Among them, biochar-based materials have the characteristics of environmental friendliness, biodegradability, low price and biocompatibility and are widely used as the supporting substrate of photocatalysts, which effectively solves the problem of photocatalyst recovery, provides a reaction surface for the catalyst and at the same time, the catalyst also improves the catalytic ability of the original biochar. Du et al. developed a super-wetting antibacterial wood mesoporous composite decorated with silver nanoparticles as a catalyst and removed organic dye pollutants (MB, RhB) from an aqueous solution. The degradation rates of MB and RhB by Ag@Wood are as high as 94.4% and 81.3%, respectively. However, the preparation of the Ag@Wood composite is expensive [57]. In the same year, Luo et al. prepared a new Z-shaped pCN/WFB/BiVO_4_ composite by a simple method and achieved high-efficiency degradation of RHB (97.3% within 30 min). The reusability of pCN/WFB/BiVO_4_ composites was also discussed, whose degradation efficiency could reach more than 90% after four cycles [58].

A better choice is to make biochar and other materials into a hybrid material as the substrate, which can combine the advantages of foreign hybrid materials and achieve better performance. For example, chitosan and phthalic anhydride were used to modify the surface of biochar, which increased its surface functional groups and had strong selective adsorption of Cu ions [59]. In addition to the modification of organic chemicals, Zhu et al. modified biochar with molybdenum disulfide (MoS_2_) to prepare a more effective adsorbent [60]. In the batch adsorption experiment, the obtained MoS2–biochar hybrid material showed good selective adsorption capacity for Pb^2+^ (189 mg g^−1^). Rui et al. used the surface plasmon resonance effect of Ag nanoparticles to improve the visible light response of TiO_2_ attached to biochar to degrade methyl orange; however, silver is relatively expensive [41].

Nowadays, with the continuous improvement of critical environmental standards, it is difficult for photocatalysts with a single load to meet the actual requirements. A woody biochar-based photocatalyst can realize multiple loads. Compared with a single-load photocatalyst, a multi-load woody biochar-based photocatalyst can degrade multiple pollutant groups simultaneously. Moreover, the composite photocatalyst will form a heterojunction, which further enhances the absorption of light to enhance the photocatalytic ability, such as the WO_3_-ZnO composite photocatalyst [12]. The existence of WO_3_ inhibits the growth of ZnO particles, increases its surface area and inhibits the recombination of photogenerated electrons. The activity of ZnO with 2% WO_3_ calcined at 600 °C is twice that of pure ZnO. In the g-C_3_N_4_-TaON composite photocatalyst [13], the internal reconstruction electric field of C_3_N_4_ and TaON drives the separation of electrons and holes in the semiconductor, which improves the performance of the catalyst. The G-C_3_N_4_/SmVO_4_ composite photocatalyst [15] and the BiOBr-g-C_3_N_4_ composite photocatalyst [61] also improved the catalytic performance of the original pure photocatalyst.

The photodegradation rate of the Mn-doped TiO_2_ structure is 96%, and its photocatalytic activity for methylene blue degradation under visible light irradiation is higher than that of undoped samples [62]. This is because the photocatalyst forms a heterojunction, changing the doping level and the electronic energy level structure, thus changing the activity of the catalyst [13]. Generally, a heterojunction is defined as the interface between two different semiconductors with unequal band structures, which may lead to an energy band arrangement [63,64] (Figure 2). Conventional heterojunction photocatalysts with staggered gaps are suitable for enhancing the separation of electron–hole pairs so that redox reactions occur on the semiconductor surface [65].

However, due to the fast electron–hole recombination, the enhancement of electron–hole separation on the heterojunction is not enough to overcome the ultra-fast electron–hole recombination and low light utilization rate on semiconductors, and the efficiency of the photocatalytic reaction is still very low. Some studies have made great efforts to solve these problems. Especially due to the spatial separation of photo-generated electron–hole pairs, a properly designed heterojunction photocatalyst is proven to have higher photocatalytic activity. Therefore, the concept of a p-n heterojunction photocatalyst is proposed. By providing an additional electric field, the electron–hole migration on the heterojunction is accelerated, thus improving the photocatalytic performance [66,67,68]. Different crystal planes on a single semiconductor can have different energy band structures. Because a heterojunction is formed by combining two semiconductor materials with different energy band structures, a heterojunction can be generated between two crystal planes of a single semiconductor, that is, a surface heterojunction. Although all the above heterojunction photocatalysts can effectively enhance electron–hole separation, the reduction and oxidation processes occur on semiconductors with lower reduction potential and oxidation potential, thus sacrificing the redox ability of photocatalysts. Z-scheme photocatalysis was proposed to maximize the redox potential of heterojunction systems [69]. Then, an all-solid-state Z-type photocatalyst was developed [70]. However, the electronic medium needed to improve the electron migration path in the all-solid-state Z-type photocatalyst is expensive and scarce, which limits the large-scale application of these photocatalysts. Later, the direct Z-type heterojunction photocatalyst did not need a rare and expensive electronic medium, and the direct Z-type photocatalyst was prepared by combining two different semiconductors without an electronic medium [71], with a low manufacturing cost. In addition, there are semiconductor–graphene photocatalysts [72,73,74]. Next, the results of other researchers’ work on biochar-based photocatalyst are summarized and compared through Table 1 [75].

## 4. Preparation Process

### 4.1. Preparation of Woody Biochar

The woody biochar-based photocatalyst is composed of a woody biochar substrate and a photocatalyst. The process is divided into the preparation of the woody substrate and the coupling of the woody substrate and the photocatalyst. Wood-based biochar is a kind of biochar, including wood bio-sponge, wood aerogel, nano-wood film and so on. Generally, biomass consists of three main components: cellulose, hemicellulose and lignin [90].

The first thing to consider is the classification of biomass raw materials used to produce biochar, which is important because the selection and feasibility of biomass pretreatment methods depend largely on the type of raw materials (wet or dry) [91]. The classification of wet biomass and dry biomass raw materials is based on the initial moisture content. A few biomasses, such as wood, usually have a low moisture content (<30%) at harvest, so they are classified as dry biomass [92]. Wet biomass can be dried into raw materials with low moisture content by supplementary drying technology, but this technology is highly energy-intensive and will reduce the overall economic benefits of the system [93,94]. On the contrary, high moisture content (>30%) is called wet biomass.

### 4.2. Carbonization Methods

Hydrothermal carbonization, also known as wet baking [95], is a thermochemical process that converts organic raw materials into high-carbon and carbon-rich solid products (corn silage [96], woody and herbal biomass [96], lignocellulose [97]). The process itself is carried out in the presence of water, so it is not affected by the high moisture content of the raw materials. The method uses water inherent in green biomass and has the advantages of non-toxicity, environmental friendliness and low price [95]. Xiao et al. used this method to increase the calorific value of corn stalk and tamarix ramosissima by 29.2 and 28.4 MJ/kg, respectively, which were 66.8% and 58.3% higher than those of raw materials [98]. Zhang et al. extracted biochar from pine and rice husk by HTC process at 300 °C, which can remove lead ions from aqueous solution, and the maximum absorption capacity is 4.25 and 2.40 mg g^−1^, respectively [99].

Dry torrefaction is a process in which biomass is heated in an inert atmosphere at a temperature of about 200–300 °C, and the residence time is 30 min to several hours [100]. This process leads to about 30% mass loss. As an important pretreatment step to improve the physical and chemical properties of biomass combustion, the roasted biomass has the characteristics of the original biomass and biochar in terms of its physical and chemical properties [91,101]. The advantages of dry torrefaction are improving energy density and grindability, achieving durability, water vapor, absorbability, dimensional stability and improving antifungal properties [102,103,104,105,106]. Sarvaramini, A. et al. roasted two kinds of woody biomass samples (birch and poplar) and their main components (cellulose, xylan and lignin) in a fixed-bed reactor at 280 °C, finding that the equilibrium moisture content decreased by about 30–40%, which indicated that the energy density was improved and the hydrophobicity was improved [107].

Gasification is the partial combustion of endogenous substances in a very high temperature range (600–1200 °C) with a short residence time (10–20 s) [53], which mainly produces hydrogen, carbon monoxide, carbon dioxide and so on, and finally obtains a small amount of biochar [108]. The average yield of typical biochar gasification is about 10 wt% of biomass [109,110]. When biomass is treated by gasification, it is found that the yield of products is influenced by fiber structure, carbon content in raw materials and organic materials; of course, moisture content also plays an important role [111]. It has been found that biomass with a low moisture content (<15 wt%) is more suitable for gasification. With the increase in moisture content, the energy demand changes proportionally [112]. Qian et al. studied the effects of three biomass types (switchgrass, sorghum and red cedar) and three equivalent ratios (0.20, 0.25 and 0.28) on the char properties obtained by gasification, which proved that the ash content of all char samples (over 40% by weight) was much higher than that of the corresponding biomass raw materials (less than 5.05% by weight) [110].

Pyrolysis is a thermochemical decomposition process in which biomass is heated at high temperatures (300–650 °C) without oxygen. The difference between pyrolysis and gasification is that there is (almost) no oxygen in the conversion process [113]. Pyrolysis mainly produces carbon-rich solid substances, namely biochar, as well as oily substances and gases such as carbon monoxide and carbon dioxide. Pyrolysis can be divided into slow pyrolysis and fast pyrolysis [114].

Slow pyrolysis, usually regarded as the main pyrolysis process, produces biochar by heating biomass at a low heating rate within a relatively long residence time (up to several days), which has a high solid yield [115,116]. Marais et al. extracted pyrolysis products from waste coal powder and recycled plastics with the help of the improved Fischer determination pyrolysis device in the Northwest University Laboratory [117], which found that the carbon produced was as high as 83% and concluded that with the increase in plastic content, the influence of temperature increase on coke output showed a linear downward trend [118]. Rapid pyrolysis produces biochar with a high heating rate (higher than 200 K min^−1^) and a short residence time (less than 10 s) [119,120]. For example, the pyrolysis of algae into biochar is such a process [121]. Wang et al. rapidly pyrolyzed microalgae in a fluidized reactor at 500 °C, and the yield of biochar was 31 wt% [122].

Biochar is formed by pretreatment, moisture control and then the pyrolysis of biomass. Slow pyrolysis is a conventional carbonization method with the longest residence time of several days and a high yield. Dry torrefaction has the lowest temperature and the longest residence time, and its product has a small loss and the characteristics of original biomass and biochar. The gasification residence time is short, but only a small amount of biochar can be obtained. Rapid pyrolysis takes the shortest time. Hydrothermal carbonization is carried out in the presence of water, so it is not necessary to consider the influence of moisture in raw materials.

### 4.3. Synthesis of Photocatalysts Supported by Woody Biochar

There are two main ways to synthesize wood-based photocatalysts: wood directly reacts with photocatalytic substances, and wood is combined with synthetic photocatalytic substances [123].

The sol-gel method is most commonly used to produce biochar-supported photocatalysts. The typical sol-gel method of biochar-supported photocatalysts follows three sequential formation steps: preparing the biochar template through thermal decomposition of biomass; increasing the surface oxide of biochar and decreasing the pH value by acid treatment; and then depositing catalytic nanoparticles on the surface of biochar. The obtained biocarbon-containing catalytic nanoparticles were calcined to obtain a stable structure. This technology forms a stable, transparent sol system through hydrolysis and condensation reactions. In the sol-gel process, there are two different stages in the synthesis process. The first stage is a colloidal suspension of particles in a liquid medium, called sol. In the second stage, the particles react with each other to form cross-linked 3-D polymer chains, which are converted into gels. Namely, there are three main steps: solvation, hydrolysis and polycondensation [123,124]. One example is that granular anatase or rutile TiO_2_ is impregnated on the surface of biochar to form TiO_2_-BSP [125,126]. On this basis, Li et al. introduced a small amount of iron ions into the TiO_2_ matrix and coated it on the surface of wood, which greatly enhanced the photocatalytic effect, stability and recycling [127]. The advantages of the sol-gel method are low equipment requirements, a simple process, diverse product forms, high purity, easy shape control and so on. In order to avoid the pollution caused by the shedding of the photocatalyst, the sol-gel method is used to make the photocatalyst evenly distributed and firmly bonded to the wooden substrate. The surface of the substrate is modified to produce a more chemically reactive surface and promote a stronger combination with the photocatalyst. It can also be considered to use a tackifier or coupling agent to enhance the combination between the photocatalyst and the sol-gel activated carbon carrier.

The hydrothermal method is a kind of liquid chemical technology that is synthesized by dissolution and recrystallization in a special closed container at high temperature and high pressure [123]. In a relatively simple process, crystalline–amorphous WO_3_-x core-shell nano-powder can be synthesized by the one-step hydrothermal method [128]. Wang et al. prepared a new photocatalyst by growing photocatalytic V_2_O_5_ in highly porous and solid DRWF and TWF substrates, which has the characteristics of economy, efficiency, sustainability and environmental protection [77]. Costa et al. reported that multistage ZnO nanostructures and microstructures were synthesized on bacterial cellulose substrates by the hydrothermal method [129]. Furthermore, Liu et al. synthesized ZnO/C composites by the microwave-assisted hydrothermal method, and the degradation efficiencies of methylene blue and rhodamine B dyes were as high as 99% and 97%, respectively, under ultraviolet irradiation [130]. The hydrothermal method has the advantages of low energy consumption and high yield, directly obtaining products with good crystallization without the need for complicated post-treatment. In addition, the hydrothermal synthesis parameters, such as temperature, pressure and duration, can be fine-tuned when synthesizing wood biochar-based photocatalysts by the hydrothermal method, which is helpful to enhance the adhesion and stability of the photocatalyst. It can also be considered to enhance the mechanical stability of wood biochar-based photocatalysts by optimizing the particle size distribution, compactness or structural integrity of materials. All these are beneficial to prevent the photocatalyst from falling off and causing pollution.

The immersion method is a method in which the matrix material is immersed in the solution of the main catalyst material and the solution of the auxiliary catalyst material, and the active substance is attached to the matrix in the form of ions or compounds [123]. Sheng et al. reported a new type of wood photocatalyst, which was deposited on the wood substrate by nano-structured Fe-doped WO_3_. When the iron content is 4.56%, the efficiency of photocatalytic degradation of formaldehyde can reach 98.21% within 6 h [131]. It is widely regarded as a simple and economical method that is suitable for various types and sizes of substrates, and the active substances are distributed on the surface of the substrate. In order to avoid the pollution caused by the shedding of the photocatalyst, a high-quality photocatalyst that can be attached to the wooden substrate should be used as much as possible during immersion, and the proper amount of photocatalyst loaded on the wooden substrate should be ensured. At the same time, handle the impregnated activated carbon carefully to prevent physical damage that may lead to the loss of photocatalyst particles.

Among these methods, the sol-gel method is the one most commonly used. This method has a simple technological process and does not need expensive instruments, but there are also some problems, such as long treatment times, high water content of the product, easy cracking due to weightlessness in the drying stage and possible pores in the product. The immersion method should wait until the impregnation reaches equilibrium before removing the impregnation solvent, which may require drying and other conditions. In contrast, the hydrothermal method can directly obtain good crystalline products without post-treatment and with a high yield. Then, the synthetic methods of photocatalysts supported by wood biochar were compared by Table 2 [132].

There are also special types of magnetic biochar composites and wooden films. Most of the original biochar is difficult to separate from wastewater because of its small volume, which limits the application of the original biochar. It is an effective method to enhance separation by transforming pure biochar into magnetic biochar. Many types of nano-metal oxides/hydroxides, such as nano-Fe_3_O_4_ and BiFeO_3_, mentioned above, not only have the functional characteristics of nanoparticles but also have unique physical and chemical properties, such as ferromagnetism and catalytic ability [140]. There are two common methods for synthesizing magnetic carbonaceous adsorbents. One is that magnetic biochar, also known as pre-saturation of biomass in an iron precursor, pyrolyzes at high temperature [141,142]; the other is chemical co-precipitation of biochar in an Fe^3+^/Fe^2+^ solution upon addition of NaOH until the pH reaches 10–11 [143,144,145].

Highly porous and flexible wood films can be obtained by removing lignin and hemicellulose from wood boards. A simple method for preparing a membrane for purifying an emulsion using biomass is introduced. Natural wood was chemically treated with a solution containing sodium hydroxide (NaOH) and sodium sulfite (Na_2_SO_3_) at 80 °C and then treated with a solution of hydrogen peroxide (H_2_O_2_) at the same temperature.

During alkali treatment, NaOH removes lignin; Na_2_SO_3_ sulfonates lignin and causes it to dissolve rapidly in the NaOH solution. At the same time, hemicellulose was also removed by the NaOH/Na_2_SO_3_ solution [145,146]. The lignin and hemicellulose components were further removed by subsequent chemical treatment with H_2_O_2_. Hydrophobic and rigid lignin and hemicellulose components were successfully removed, while soft and hydrophilic cellulose remained. Due to the removal of hydrophobic lignin, the obtained wood film showed excellent water absorption and underwater oil adhesion resistance [147].

When synthesizing photocatalysts based on wood biochar, it is necessary to maintain the integrity and properties of biochar and add semiconductor photocatalysts to enhance photocatalytic activity [148,149]. Here are some strategies to keep biochar intact during synthesis: (1) Minimize aggressive treatment: minimize harsh chemical or physical treatment of wood biochar to prevent its structure and characteristics from being changed. For example, choose mild chemicals and solvents in the synthesis process, avoid strong acids and alkali, remove impurities and wash with water or a weak acid [150]. (2) Control the temperature and time of synthesis: keep the best synthesis conditions, including temperature and reaction time, to prevent excessive degradation of wood biochar [150]. High temperatures or prolonged reaction times will lead to the destruction of the wood biochar structure. (3) Optimization of synthesis parameters: fine-tune synthesis parameters, such as precursor concentration, pH value and mixing conditions, to ensure the effective combination of the photocatalyst while maintaining the characteristics of wood biochar [151,152]. (4) Surface modification technology: using surface modification technology, a semiconductor photocatalyst is selectively deposited on the surface of biochar without affecting the bulk structure. Techniques such as immersion or sedimentation are helpful to maintain the integrity of biochar [153]. (5) Characterization and evaluation: whether the structure and properties of wood biochar have changed can be judged by characterizing the synthesized wood biochar-based photocatalyst. The composites can be analyzed by SEM, TEM, XRD and FTIR [154,155].

## 5. Limitation of Woody Biochar-Based Photocatalysts in Water Treatment

One limitation of photocatalysts based on wood biochar in the field of water treatment is that, compared with other semiconductor-based photocatalysts, their efficiency in degrading some types of pollutants or organic compounds in water is relatively low [156]. This may lead to the slow or incomplete removal of pollutants, thus reducing the water treatment effect.

The selectivity of wood biochar-based photocatalysts to specific pollutants may limit their applicability in treating different water sources with different pollutant compositions [157]. This selectivity will limit their effectiveness in solving various water pollution problems, especially those involving complex pollutant mixtures.

The stability and reusability of wood biochar-based photocatalysts in water treatment applications are also problems [158]. The leaching of active components, the physical degradation of the biochar matrix and the loss of photocatalytic activity after repeated recycling will affect the long-term performance and cost-effectiveness of these materials in the process of continuous water treatment.

To solve these limitations, further research and development work may be needed to optimize the design, synthesis and performance of wood biochar-based photocatalysts specially used for water treatment applications. This may involve enhancing its photocatalytic activity, improving its stability and reusability and exploring strategies to expand its applicability in order to effectively treat a wider range of pollutants in water systems.

In addition, a shadowing effect may appear in the photocatalytic reaction of a wood biochar-based photocatalyst [159,160]. This phenomenon is usually due to the fact that the light cannot fully penetrate the thick or dense part of the photocatalyst, thus reducing the effective light absorption on the surface of the catalyst, resulting in some areas not being irradiated by light and affecting the catalytic reaction [161,162,163]. The shadowing effect can be reduced by the following strategies: (1) Adjusting the structure of the catalyst [164]: by designing the catalyst structure with a larger pore size and lower density, the penetration depth of light can be increased and the shadowing effect can be reduced. (2) Surface modification [165]: the light absorption efficiency can be improved by surface modification of wood biochar-based photocatalysts, such as changing the surface chemical properties and introducing active sites of the catalyst. (3) Porous structure design [166]: the design of a photocatalyst with a porous structure can increase the reaction surface area, improve the light absorption efficiency and reduce the shadowing effect. (4) Angle control of the light source [166]: adjusting the angle and position of the light source so that the light can irradiate the catalyst surface more evenly.

## 6. Summary and Outlook

Wood-based photocatalysts have been used as carrier substrates for photocatalysts at macro- and micro-levels. As a composite whole, they are jointly used in sewage treatment. The research on photocatalysts has made great achievements and played a great role in sewage treatment. However, it is difficult to recycle, and even some of them are toxic. Woody biochar is economical and easy to obtain, and it also has the ability to treat sewage, but it is relatively weak. The advantages of combining the two complement each other, and the disadvantages cancel each other out. Therefore, an efficient and economical woody biomass substrate appears, which is combined with a photocatalyst and used in sewage treatment at the same time. For example, in terms of treatment efficiency, the treatment efficiency of a suitable composite photocatalyst under visible light is greater than that of a single catalyst, and the substrate can be made magnetic, which is more conducive to the recovery of the catalyst. Looking forward to the future, the development of this matrix catalyst system from a wood composite photocatalyst seems promising. The adaptability of the wood biochar matrix is used to change the macrostructure, which provides the possibility of customized shape and design improvement, thus further enhancing the catalytic performance. It is expected to be commercialized to develop a cost-effective manufacturing process for wood biochar-based photocatalysts and ensure the scalability of the production method. The combination of a photocatalyst based on wood biochar with other advanced technologies (such as sensor systems, artificial intelligence or nanotechnology) can further enhance its performance and applicability. At the same time, it is very important to continue to work hard to improve the environmental sustainability of wood biochar photocatalysts. This includes optimizing the recycling process, minimizing energy consumption in the production process and ensuring that the overall life cycle impact of these materials is as environmentally friendly as possible. Generally speaking, wood biochar-based photocatalysts show great application potential in sewage treatment and other environmental remediation. Continuous research and innovation in this field will probably further promote the development of efficient and environmentally friendly catalyst systems.

## Figures and Tables

**Figure 1 ijms-25-04743-f001:**
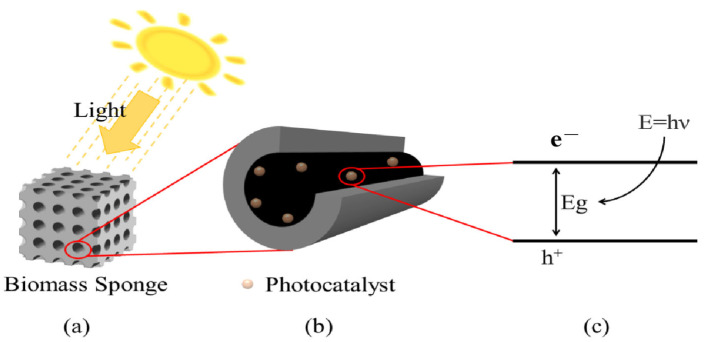
(**a**) is the abstract structure of the supported photocatalyst. (**b**) is a graph of photocatalyst particles loaded on the pores of the substrate. (**c**) is an electron excited by a photocatalyst. The relationship from a to c is gradually enlarged.

**Figure 2 ijms-25-04743-f002:**
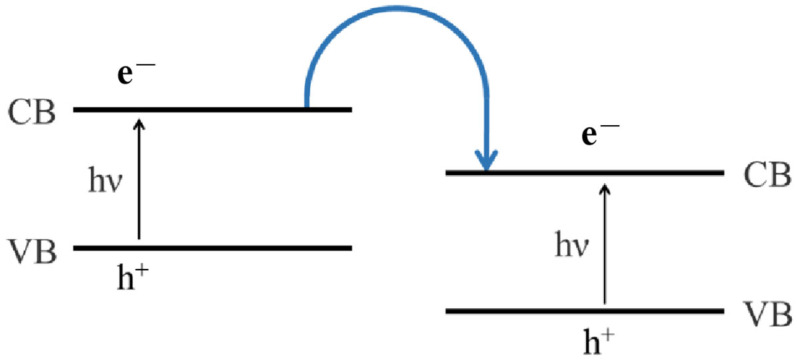
Schematic diagram of heterojunction energy level.

**Table 1 ijms-25-04743-t001:** Photocatalytic efficiency and recycling cycle of different wooden carbon-based photocatalysts.

Wood Species	Photocatalyst	Pollutant	PH	Light Source	Efficiency	Cycles of Reuse	Ref
Balsa wood	BiOBr/AgB	Rhodamine B (5 mg L^−1^)	4	500-W xenon lamp	99%	5	[75]
Fir wood	CdS quantum dots/nano-TiO_2_	Co (20 mg L^−1^)	2	High-pressure mercury lamp	Almost 100% in 80 min	83.4% after 5 cycles	[76]
Balsa wood	V_2_O_5_ nanowires	Methylene Blue (100 mg L^−1^)	4.6	500-W xenon lamp	51.6%	10	[77]
Carbonized wood	AgI/UiO-66-NH_2_	Rhodamine B (10 mg L^−1^)	Not reported	300-W xenon lamp	96%	Above 85% after 6 cycles	[78]
Basswood	Pt/Au/TiO_2_	Co (40 mg L^−1^)	Not reported	300-W xenon lamp	94% in 80 min	Not reported	[79]
Beech wood	Fe_3_O_4_	TOC (19.4 mg L^−1^)	1–9	100-W mimetic solar irradiation	51% in 40 min	4	[80]
Balsa wood	TiO_2_	Methylene Blue (0.1 mmol L^−1^)	Not reported	Sunlight	97.7% in 12 h	Not reported	[81]
Poplar wood	CQDs@ MIL-88B	Methylene Blue (10 mg L^−1^)	2	450-W xenon lamp	98% in 60 min	92% after 5 cycles	[82]
Natural wood	UiO-66	OFX (4 ppm)	2–8	Simulated sunlight	80.96%	4	[83]
Honeylocust spine wood	TiO_2_	BPA (20 mg L^−1^)	7	20-W UV lamp	80.08%	Not reported	[84]
Walnut shell	Ag/TiO_2_	MO (20 mg L^−1^)	Not reported	500-W mercury-vapor lamp	97.48% in 60 min	96.10% after 5 cycles	[41]
Carbonized wood	Ag_3_PO_4_	Methylene Blue (5 ppm)	4.6	Sunlight	About 65% in 90 min	Not reported	[85]
Balsa wood	Cu_2_O	Methyl orange (20 mg L^−1^)	4	Simulated visible light	94.5% in 140 min	Above 80% after 5 cycles	[38]
White oak	TiO_2_	Rhodamine B (1 × 10^−5^ M)	Not reported	1 kW cm^−2^ irradiation	92% in 60 min	85% after 6 cycles	[86]
Wood waste	ZnO/AC	Methylene Blue (100 mg L^−1^)	8	Visible light	89.73% in 180 min	Not reported	[87]
Wood flour	BiOBr	Rhodamine B (20 mg L^−1^)	Not reported	200-W LED	Almost 100% in 30 min	Not reported	[88]
Poplar wood	pCN/BiVO_4_	Rhodamine B (10 mg L^−1^)	3	400-W mercury lamp	97.3% in 30 min	Above 90% after 4 cycles	[58]
Wood-based activated carbon fiber	Mn/TiO_2_	Methylene Blue (33 mg L^−1^)	Not reported	Visible light	96% in 4 h	Not reported	[62]
Balsa wood	Ag	MB/NaBH_4_ (100 mg L^−1^)	2–12	Simulated visible light	94.45% in 10 h	10	[57]
Fir wood	Cu_2_O	Methylene Blue (6 mg L^−1^)	5	Visible light	92.82% in 12 h	Above 80% after 60 cycles	[89]

**Table 2 ijms-25-04743-t002:** Comparison of synthesis methods of photocatalysts supported by wood biochar.

Synthetic Method	Operational Approach	Advantages	Limitations	Ref
Sol-gel method	(1) Solvation; (2) Hydrolysis; (3) Polycondensation.	High purity; Low processing temperature; Good adhesion; Low equipment requirements; Simple process; The reaction is easy to control; Product forms are diverse; Homogeneous coating.	Longer time requirements; The manufacturing cost is higher; High water content; The drying stage is easy to crack due to weightlessness; There may be pores; Potential agglomeration; Limited scalability.	[132,133,134]
Hydrothermal method	(1) Selecting a reaction precursor and determining the ratio of the precursors; (2) Determining the adding sequence of precursors and stirring wet materials; (3) Load and seal the autoclave, and then put it into the oven; (4) Select the reaction temperature, time and state.	Enhanced crystallinity; The grain linearity is moderately adjustable; No complicated post-treatment; Low reaction energy consumption; High yield; The crystal form and morphology are related to hydrothermal conditions.	Limited precursor options; Limited scalability; Precise control of reaction parameters; By-product formation; Limited application range.	[135,136,137]
Immersion method	(1) Preparation of semiconductor support; (2) Impregnation of precursor; (3) Drying; (4) Thermal treatment.	Suitable for substrates of various shapes and sizes; The forming steps are quick and simple; High utilization rate; Relatively low cost.	Limited Penetration; Potential agglomeration; Difficulties in controlling nanoscale features; Precursor solubility; Drying and other conditions may be required.	[138,139]

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
