# Peer review of "A Wooden Carbon-Based Photocatalyst for Water Treatment"

_ijms, 2024, doi:10.3390/ijms25094743_

Round 1
Reviewer 1 Report
Comments and Suggestions for Authors
Comments and Suggestions for Authors
For the review article entitled (Woody Biochar-based Photocatalyst for Water Treatment).
The authors have done interesting work through the review article , but I have some comments to be addressed.
1. The review article title should be improved and the word woody should be changed.
2. The authors should improve the background about Biochar from wood source and other Biochar sources for photocatalytic and adsorption applications.
3. The authors should add a table containing the work conducted by other researchers and their work output listed with references.
4. The authors should add some references expressing the different sources of Biochar and carbon-based materials for both photocatalytic activity and adsorption.
Reviewer 2 Report
Comments and Suggestions for Authors
Woody Biochar-based Photocatalyst for Water Treatment
This paper could be accepted after minor revision.
1. Line 50- pyrolysis temperature could attain up to 950°C to form biochar.
2. Lines 74-80- the authors do not discuss woody biochar-based photocatalyst in this part.
3. Line 85- Hybridization is not the correct term here. Please use the formation of a hybrid material.
4. Hydrochar is not a biochar. Therefore, the title of the paper is not correct. Please use the title: “Woody carbon-based Photocatalyst for Water Treatment”.
Comments on the Quality of English LanguageSmall revision
Reviewer 3 Report
Comments and Suggestions for Authors
The authors present an interesting review, however a bit of rearranging is needed to make the review appeal to many readers and ensure the information is easily transferable
Introduction
The last paragraph of the intro in the review, let go of “We”, also throughout the document
Also the last sentence of 6 lines is too long and hard to read, break it up into at least 2 sentences
Subsections
Section 2 should be changed to Section 3
The in Section 2, Authors need to talk about what Biochar is, the properties, different types and particualry focusing on the optical properties as that is what will make the material suitable as a photocatalyst. Also touch on the surface area and particle size
Then 3rd section, authors need to talk about the synthesis-remain as is but add a table on this section, highlighting the different methods, how they operate, advantages and limitations
Section 4: Woody biochar photocatalyst
A Table showing biochar based photocatalysts
It needs to include: Material, dosage, Ph, % degradation, Reff – Comparison table
-Lastly, the authors need to add a section on the limitation of wood biochar photocatalysts
-Summary and Outlooks need to be elaborated on
Comments on the Quality of English LanguageThe authors needs to proof read the manuscript again even after making the changes to ensure minimal gramatical errors
Reviewer 4 Report
Comments and Suggestions for Authors
This paper combines photocatalyst with biochar for water treatment. The mechanisms about the processes of water treatment by biochar-based photocatalyst were investigated by several characterization techniques. The studies were quite systematic and the resulted were well organized by the authors. I’d like to recommend the publication of this paper in Intermational Journal of Molecular Sciences (IJMS) after revision.
1. The method about photocatalyst loads on charcoal should be discussed for making sure if photocatalyst falling off will produce pollution by authors.
2. The phenomenon of shadowing effect on woody biochar-based photocatalyst should be discussed for water treatment for understand if photodegradation efficiency was affected in the reaction by authors.
3. The duration and re-use of woody biochar-based photocatalyst should be discussed for long term application.
Round 2
Reviewer 1 Report
Comments and Suggestions for Authors
Accept in present form
Reviewer 3 Report
Comments and Suggestions for Authors
The authors have greatly improved the manuscript with the suggested additions/revisions and the work can be accepted for publication